# Mental Health Shame, Caregiver Identity, and Self-Compassion in UK Education Students

**DOI:** 10.3390/healthcare10030584

**Published:** 2022-03-21

**Authors:** Yasuhiro Kotera, Freya Tsuda-McCaie, Ann-Marie Edwards, Divya Bhandari, Dan Williams, Siobhan Neary

**Affiliations:** 1School of Health Sciences, University of Nottingham, Nottingham NG7 2HA, UK; 2College of Health, Psychology of Social Care, University of Derby, Derby DE22 1GB, UK; freya.mccaie@gmail.com (F.T.-M.); annm.edwards@icloud.com (A.-M.E.); 3Medical Governance Research Institute, Tokyo 108-0074, Japan; rayordeal3@gmail.com; 4College of Arts, Humanities and Education, University of Derby, Derby DE22 1GB, UK; d.williams@derby.ac.uk (D.W.); s.neary@derby.ac.uk (S.N.)

**Keywords:** education students, mental health, mental health shame, caregiver identity, self-compassion

## Abstract

Although students in education have high rates of mental health problems, many of them do not ask for help, which can exacerbate their symptoms. One reason for their low help-seeking is shame associated with mental health problems. As education students aspire to provide care for children, they may feel ashamed to care for themselves, as the role identity theory suggests. Self-compassion is reported to reduce shame and mental health problems. This study explored the relationships between mental health problems, mental health shame, self-compassion, and caregiver identity among UK education students. One hundred and nine postgraduate students completed four self-report scales regarding those constructs. Correlation and regression analyses were performed. Mental health problems were positively associated with shame and identity, and negatively associated with self-compassion. Self-compassion was the only significant predictor of mental health problems. Findings will help educators and education students to develop effective approaches for their mental health problems.

## 1. Introduction

### 1.1. Mental Health of UK Postgraduate Education Students

Mental health problems among postgraduate students in the UK are concerningly prevalent [1]. Globally, graduate students are six times more likely to experience anxiety and depression than the general population [2]. Postgraduate education students, including trainee teachers, may be especially vulnerable to mental health problems, because of the stress and workload associated with managing work placements and academic work [3]. Additionally, the emotional impact of undergoing teacher training is significant, and students experience anxiety, self-doubt, and fatalistic feelings [4]. Indeed, psychological distress is prevalent among trainee teachers in the UK [5]. Poor mental health is associated with reduced academic achievement and attrition in tertiary education [6,7]. Despite this, research on the mental health of education postgraduate students is scarce [5], and no studies have investigated the relationships between mental health shame, caregiver identity, and self-compassion on mental health symptoms in this population.

### 1.2. Role Identity

Role identity has been used to explore helping professionals’ reluctance to seek help for mental health problems. Role identity theory argues that individuals’ perceptions of their personal and social identities, and societal and individual associations with those identities, shape role identity. Role identity is an idealised version of self, which provides a schema for appraisal and determination of thoughts and action, and creates expectations of self [8]. Siebert and Siebert [9] applied role identity theory to helping professionals’ reluctance to seek help for mental health issues. They theorised that the association between caregiver identity and lack of help-seeking may occur because social workers overlook their mental health problems because of an unwillingness to acknowledge that they have similar problems to their clients, as such an acknowledgment would challenge and destabilise their idealised role identity [9]. A caregiver identity is viewing oneself and/or believing others view one as a helper or caregiver either professionally or personally (or both). Role identity as a caregiver was associated with burnout, depression, and not seeking help among social workers [9], and, in healthcare students, was predictive of mental health problems [10,11,12].

Caregiver identity may be prevalent among education students, and exacerbate mental health challenges by decreasing help-seeking behaviour. Education students’ social identities, influenced by their professional and academic roles, may include the identity of ‘teacher’ [13]. Given the responsibilities that comprise teachers’ work (i.e., nurturing, safeguarding, and developing young people), postgraduate education students may associate the role of ‘teacher’ with ‘caregiver’, and thus identify as caregivers. Indeed, teaching is classified as one of the caring professions [14]. Furthermore, popular social discourses depict ‘good’ teachers as selfless, heroic [15], and caring [16]; that is, ‘good’ teachers foreground and care for their pupils’ wellbeing. These associations with the teacher identity may impact education students’ expectations of themselves in the same way that idealised notions of ‘helping professionals’ impacted social workers’ self-expectations [9]. Thus, teachers may face difficulties in reconciling their role with that of someone in need of support, thereby inhibiting acknowledgment of, and seeking help for, mental health problems. Indeed, the British Education Research Association argue that that challenges around the teacher identity may contribute to mental health problems among postgraduate education students [3]. Yet, caregiver identity and its relationship with mental health has not been explored with postgraduate education students. 

### 1.3. Mental Health Shame

Mental health shame is feeling ashamed for having a mental health problem (Kotera et al. 2019d), and has been associated with poor mental health in business, psychotherapy, occupational therapy, social work, and nursing students in the UK [10,11,17,18]. One explanation for this association is that mental health shame reduces the likelihood that an individual will seek help for their mental health problems [19]. Indeed, mental health shame was associated with reduced help-seeking among young adults with mental health problems [20], and UK social work students [21]. 

Mental health shame may be especially prevalent among postgraduate education students because of incompatibility between stigmatised notions of people with mental health problems and idealised depictions of educators. Stigmatised attitudes may be internalised, and result in feelings of shame [22]. Stigmatised beliefs about people with mental health problems include that they are weak, irresponsible, childlike, and unable to care for themselves [23]. Identities associated with teachers include ‘expert’ and ‘authority’ [24], and ‘manager’ or ‘professional’ [25]. These two identities may seem incompatible. Additionally, within the teaching profession, the dominant morality foregrounds institutional wellbeing over individual wellbeing, promoting resilience as the solution to work-based challenges [26]. Trainee teachers may internalise this morality, and interpret mental health problems as reflecting a lack of personal resilience. Thus, they may feel their mental health problems are indicative of weakness and their poor suitability as an educator—exacerbating mental health shame, and preventing help-seeking. Relatedly, caregiver identity may heighten mental health shame. Indeed, among social work students, mental health shame was associated with caregiver identity [10]. 

### 1.4. Self-Compassion

Self-compassion is associated with improved wellbeing and decreased mental health problems [27]. The construct has three central components: self-kindness, common humanity, and mindfulness [28]. Self-kindness describes an understanding and sympathetic attitude towards oneself in response to failure or pain (rather than self-criticism and judgment). Common humanity is the feeling that one’s experiences, including painful or difficult ones, are shared aspects of the human condition, and thus are unifying rather than isolating. Mindfulness is a state in which painful thoughts and feelings can be acknowledged and held, but not overly identified with or equated with the self [28]. Neff posits that self-compassion conceptualises a healthy relationship to oneself, and is protective against the negative impacts of rumination, self-judgement, and isolation [28]. Research supports the protective mechanism of self-compassion, and it has been negatively associated with mental health symptoms among UK social work, counselling, occupational therapy, and business students [10,11,12,17] with burnout; reduced wellbeing and self-criticism among UK midwifery students [29]; and with lower distress among university students more broadly [30]. 

### 1.5. Three Affect Regulation Systems

Gilbert theories that imbalances between the three affect regulation systems can lead to mental health problems. The three systems regulating affect are: (i) the threat and protection system; (ii) the drive, excitement, and resource-seeking system; and (iii) the contentment, soothing, and safety system [31]. Imbalances—especially an overdeveloped and dominant threat system, and an under-activated soothing system—are a source of psychopathology and poor wellbeing [31]. Individuals with high levels of shame and self-criticism have dominant threat systems, and struggle to activate their soothing systems [32]. Thus, self-criticism and shame, in combination with low levels of self-reassurance, leads to an activated threat system, resulting in lowered wellbeing and an increased risk of mental health problems [31]. In contrast, self-compassion activates the soothing system, creating balance between the three systems [31]. Indeed, compassionate mind training reduced participants’ levels of self-criticism and shame [33].

The interplay of affect regulation systems may explain the protective role of self-compassion, and the negative impact of caregiver identity and mental health shame on mental health problems. Mental health shame may result in an activated threat system, increasing risk of mental health problems. Caregiver identity, being positively associated with mental health shame, may also result in an activated threat system. Additionally, caregiver identity may result in heightened self-criticism (associated with the threat system), as individuals feel a sense of failure or inadequacy around experiencing mental health challenges. In activating the soothing system, self-compassion may counter the negative impacts of mental health shame and caregiver identity on mental health problems. However, no study has investigated the relationship between self-compassion, mental health shame, and mental health problems in postgraduate education students. Understanding the prevalence and functions of caregiver identity, self-compassion, and mental health shame in the development of mental health problems among postgraduate education students may help universities and health services develop targeted, ameliorative strategies to prevent or address mental health problems, and improve students’ wellbeing.

### 1.6. Aims

Therefore, this study aimed to explore relationships among mental health problems, mental health shame, self-compassion, and caregiver identity in UK education students. Mental health was evaluated in terms of depression, anxiety, and stress, as anxiety and depression are the most common mental health disorders in the general population [34], and stress is common among trainee teachers [5]. Three research questions were considered:
RQ1. How are mental health shame, self-compassion, and caregiver identity related to each mental health problem (depression, anxiety, and stress)?RQ2. How do mental health shame, self-compassion, and caregiver identity predict each mental health problem (depression, anxiety, and stress)?RQ3. How do mental health shame and caregiver identity predict self-compassion?


## 2. Materials and Methods

### 2.1. Participants

Participants had to be 18 years old or older, and studying in an education programme at a UK university at the time of the study: students taking a study break were excluded. Participants were recruited via convenience sampling using a paper-based survey distributed by programme tutors instead of the researchers to avoid response biases. Of 120 part-time graduate students who were introduced to the study, 109 (91%; 70 females, 39 males; Age 27.39 ± 7.94 years old, range 21–55 years old; 104 British, 2 other Europeans, and 1 Asian) completed four mental health scales, satisfying the required sample size calculated by power analysis (84: two tails, *p* H1 = 0.30, α = 0.05, Power = 0.80, *p* H0 = 0) [35]. Coloured paper was prepared for students with visual impairments, and one student used it. Paper data were digitised by a research assistant, who was not a co-author of this study. Compared with the general population of UK education students, which is 78% [36], our sample recruited slightly fewer females (64%). No compensation was awarded for completing the survey. We did not ask for a reason for withdrawal to the 11 participants, adhering to the ethical guidelines: no reason nor complaint was received. Ethics approval was obtained from the University Research Ethics Committee.

### 2.2. Instruments

The Depression Anxiety and Stress Scale (DASS21), a shortened version of DASS42 [37], was used to assess mental health problems. DASS21 consists of 21 items on a four-point Likert scale (0 = did not apply to me at all to 3 = applied to me very much or most of the time; a higher score indicates poorer mental health) divided into three subscales, seven items each: depression (e.g., ‘I found it difficult to work up the initiative to do things’), anxiety (e.g., ‘I was worried about situations in which I might panic and make a fool of myself’), and stress (e.g., ‘I found myself getting agitated’). These subscales had good reliability: α = 0.87–0.94 [38].

Mental health shame was assessed using the Attitudes Towards Mental Health Problems (ATMHP), consisting of 35 four-point Likert items (0 = do not agree at all to 3 = completely agree; a higher score indicates stronger shame). ATMHP involves four sections: (i) their community’s and family’s attitudes towards mental health problems (community and family attitudes, e.g., ‘My community/family sees mental health problems as a personal weakness’); (ii) their views on how their community and family would see them if they had a mental health problem (community and family external shame, e.g., ‘I think my community/family would see me as inferior’); (iii) how they view themselves if they had a mental health problem (internal shame, e.g., ‘I would see myself as inadequate’); and (iv) how their family would be perceived if they had a mental health problem (family-reflected shame, e.g., ‘My family would be seen as inadequate’), and how much they worry about themselves when a close relative had a mental health problem (self-reflected shame, e.g., ‘I would worry that others would not wish to be associated with me’). All of the subscales had good Cronbach’s alphas of between 0.85 and 0.97 [39].

Self-compassion was evaluated using the Self-Compassion Scale-Short Form [40]. This self-report measure is a shortened version of the original 26-item Self-Compassion Scale [28], comprising 12 items (e.g., ‘When something painful happens I try to take a balanced view of the situation.’) on a five-point Likert scale (1 = almost never to 5 = almost always; a higher score indicates more self-compassion). Cronbach’s alpha was found to be 0.86 [40].

Lastly, the Role Identity Scale (RIS) was used to assess caregiver identity. This eight-item measure considers (a) how participants view themselves as a caregiver, and (b) how they perceive how others view themselves as a caregiver [9]. Participants respond to how much they agree to each item (e.g., ‘I have heard I am a natural helper or caregiver’) on a five-point Likert scale (1 = strongly disagree to 5 = strongly agree; a higher store indicates a stronger caregiver identity). The reliability of RIS was high (α = 0.78). 

### 2.3. Procedure

First the collected data were screened for outliers and the assumptions of parametric tests. Second, correlations between their mental health problems, mental health shame, self-compassion, and caregiver identity were examined. Third, multiple regression analyses were conducted to identify significant predictors for each mental health problem. Finally, another multiple regression analysis was conducted to identify predictors for self-compassion. 

## 3. Results

Analyses were conducted using IBM SPSS version 25.0. No outliers were identified. All variables demonstrated good internal reliability (*α* = 0.87–0.96; Table 1). 

### 3.1. Relationships among Mental Health, Mental Health Shame, Self-Compassion, and Sleep

Because all variables apart from self-compassion were not normally distributed (Shapiro–Wilk test, *p* < 0.05), data were square-root-transformed to satisfy the assumption of normality [41]. Pearson’s correlation was calculated (Table 2). 

Mental health problems were positively associated with all the mental health shame subscales (external shame being the strongest positive correlate) and caregiver identity, and negatively associated with self-compassion (RQ1). Furthermore, self-compassion was negatively related to all the mental health shame subscales (again, external shame was the strongest) and caregiver identity. 

### 3.2. Predictors of Mental Health Problems

Multiple regression analyses were conducted to explore the relative contribution of mental health shame, self-compassion, and caregiver identity to each mental health problem (Table 3). Mental health shame was calculated by summing all of the subscale scores [11]. First, gender and age were entered to statistically adjust for their effects (step one), and then mental health shame, self-compassion, and sleep were entered (step two). Adjusted coefficients of determination (Adj. R^2^) were reported. Multicollinearity was not a concern (VIF < 10). Mental health shame, self-compassion, and caregiver identity accounted for 26–49% of the variance in each mental health problem, indicating a large effect size [42]. Self-compassion was the only significant predictor (*p* < 0.001), negatively predicting all mental health problems (RQ2): Depression B = −4.22, Anxiety B = −2.99, and Stress B = −3.22. 

### 3.3. Predictors of Self-Compassion

Lastly, another multiple regression analysis was performed to appraise the relative contribution of mental health shame and caregiver identity to self-compassion (Table 4). Again, multicollinearity was not a concern (VIF < 10). Mental health shame and caregiver identity accounted for 25% of the variance in self-compassion, indicating a large effect size [42]. External shame (*p* < 0.05, B = −0.04) and internal shame (*p* < 0.001, B = −0.07) were significant predictors for self-compassion (RQ3).

## 4. Discussion

There is little evidence in the literature on the mental health of postgraduate education students, despite a prevalence of mental health problems in this particular group [43]. To the best of our knowledge, this is the first study investigating the relationship between self-compassion, caregiver identity, mental health shame, and mental health problems in postgraduate education students. Regarding RQ1, all mental health problems were indeed positively associated with mental health shame and caregiver identity, and negatively associated with self-compassion. In addition, external shame was the strongest correlate with self-compassion. Moreover, our regression analysis revealed that all mental health problems were only predicted by self-compassion (RQ2). Lastly, another regression identified that external and internal shame were significant predictors of self-compassion (RQ3). We discuss each finding in detail below.

In the present study, a positive relationship between self-compassion, mental health shame, and caregiver identity were noted, supported by previous findings showing the impact of self-compassion and caregiver identity on mental health problems [44,45]. The findings of this study demonstrate that self-compassion can result in long-lasting improvements in mental health symptoms, such as decreased depression and greater happiness [46], which would be particularly helpful for postgraduate education students, such as trainee teachers. Siebert and Siebert [9] were the first to research role identity and distress. They found that strong caregiver role identity is significantly associated with psychological and occupational distress, and negatively correlated with help-seeking [9]. These findings are consistent with [47], who highlight the importance of cultivating self-compassion in order to reduce the negative consequences associated with external shame. This has implications for the role of self-compassion-based interventions in shame reduction, which warrants further investigation [48]. The results also demonstrate the need to understand the specific challenges faced by postgraduate education students, and how practising self-compassion can prove beneficial. Counselling and other mental health care interventions may be necessary to help improve students’ health and wellbeing [49].

Though mental health shame, self-compassion, and caregiver identity predicted 26–49% of variance in each mental health problem, analyses found that self-compassion was a significant predictor, negatively predicting each mental health problem. Findings indicate that individuals with higher levels of self-compassion may be more resilient [50], thus lowering levels of anxiety, depression, and stress-related illnesses [51,52]. However, our study found that external and internal shame negatively predicted self-compassion in postgraduate education students, which is consistent with previous research on self-compassion and mental health shame [11,18]. Shame is linked to psychopathology in various ways, evoking negative feelings across a range of mental health issues, such as anxiety and depression [48,53]. Education students could benefit from Compassionate Mind Training (CMT), which has proven effective in increasing levels of self-compassion in caring professions, such as psychotherapy [29] and education [54]. Developed by Gilbert [31], CMT is a therapeutic approach designed to promote and develop self-compassion for self and others, and is regarded as an important tool to improve mood and wellbeing [46]. For example, 31 nursing, social work, counselling, psychology, and teaching students were tested to see if Mindfulness-Based Stress Reduction (MBSR) could help them avoid stress and increase self-compassion. It was found that practising mindfulness increased self-compassion levels, implying the importance of raising awareness of self-compassion, and its benefits for the mental health of students [55,56]. Indeed, mindfulness is recommended for trainee teachers [57]; however, self-compassion has not been assessed in depth in this population. The findings have implications for further research on self-compassion and its relationship with mental health problems, particularly in postgraduate education students. Moreover, incorporating wellness programmes into the university student experience [58], as well as mindfulness-based intervention in the graduate curriculum, may yield mental health benefits [59], such as enhancing the cognitive and emotional capacity of trainees to cope with the demands of caring professions [60]. Wellbeing education was effective for reducing stress among students [61]. Furthermore, if proven to be effective in these contexts, these interventions could be beneficial in other occupational and international settings. For example, how ‘self-compassion’ is perceived is different by culture and sector [62]. The ability to improve psychological wellbeing following self-compassion training is both pragmatic and beneficial; hence, it is worthy of wider applications. Further research is therefore warranted.

### Limitations

The present study has several limitations. First, the lack of ethnically diverse students is likely to limit the generalisability of those underrepresented groups. Moreover, this study did not examine the mental health of postgraduate students in other countries. Considering the cross-cultural differences observed in mental health problems and shame [63], the international generalisability of our findings needs to be assessed. Second, this study relied on self-reported measures. Results are often influenced by self-reported surveys when individuals report their own experiences, thus limiting the accuracy of the findings. Alternative data collection methods would enhance future research. Lastly, because participants were recruited from a single UK academic institution, the findings may not represent other higher education institutions, which may have implications for the mental health of enrolled postgraduate students. 

## 5. Conclusions

The findings of our study revealed that mental health problems were positively associated with mental health shame and caregiver identity. Similarly, those experiencing internal and external mental health shame were less likely to have self-compassion. However, students with higher self-compassion were found less likely to face mental health problems, which highlights self-compassion as a protective factor against mental health problems. Therefore, interventions (such as CMT and MSBR) aimed to improve self-compassion, and reduce mental health shame, should be designed and promoted to enhance the overall wellbeing of the students.

## Figures and Tables

**Table 1 healthcare-10-00584-t001:** Descriptive statistics: mental health problems, mental health shame, self-compassion, and caregiver identity in UK education students (*n* = 109).

Scale	Construct (Range)	M	SD	*α*
Depression Anxiety and Stress Scale-21	Mental Health Problems			
	Depression (0–42)	13.72	11.92	0.93
	Anxiety (0–42)	12.07	9.13	0.77
	Stress (0–42)	19.88	10.52	0.84
Attitudes Towards Mental Health Problems	Mental Health Shame			
	Negative Mental Health Attitudes (0–24)	5.20	5.36	0.91
	External Shame (0–30)	7.22	8.36	0.96
	Internal Shame (0–15)	8.68	5.24	0.97
	Reflective Shame (0–36)	7.17	7.72	0.90
Self-Compassion Scale-Short Form	Self-Compassion (1–5)	2.73	0.77	0.86
Role Identity Scale	Caregiver Identity (8–40)	30.13	5.11	0.75

**Table 2 healthcare-10-00584-t002:** Correlations among mental health problems, mental health shame, self-compassion, and caregiver identity in UK education students (*n* = 109).

		1	2	3	4	5	6	7	8	9	10	11
1	Gender (0 = M, 1 = F)	-										
2	Age	0.06	-									
3	Depression	0.11	0.06	-								
4	Anxiety	0.19 *	0.02	0.69 **	-							
5	Stress	0.23 *	0.04	0.74 **	0.70 **	-						
6	Negative Mental Health Attitudes	0.06	0.15	0.38 **	0.25 **	0.41 **	-					
7	External Shame	0.20 *	0.10	0.49 **	0.38 **	0.51 **	0.72 **	-				
8	Internal Shame	0.08	0.06	0.40 **	0.28 **	0.39 **	0.41 **	0.53 **	-			
9	Reflected Shame	0.03	0.01	0.34 **	0.24 *	0.34 **	0.48 **	0.59 **	0.62 **	-		
10	Self-Compassion	−0.13	−0.03	−0.66 **	−0.53 **	−0.66 **	−0.37 **	−0.46 **	−0.46 **	−0.27 **	-	
11	Caregiver Identity	0.20 *	0.06	0.29 **	0.26 **	0.23 *	0.22 *	0.28 **	0.32 **	0.19 *	−0.24 *	-

* *p* < 0.05, ** *p* < 0.01. For gender (0 = M, 1 = F), point-biserial coefficients are reported.

**Table 3 healthcare-10-00584-t003:** Multiple regression: mental health shame, self-compassion, and caregiver identity to mental health problems among education students (*n* = 109).

	Depression			Anxiety			Stress		
		95% CI			95% CI			95% CI	
	B	Lower	Upper	B	Lower	Upper	B	Lower	Upper
Step 1									
Gender (0 = M, 1 = F)	0.39	−0.34	1.13	0.63 *	0.02	1.24	0.69 *	0.13	1.26
Age	0.01	−0.03	0.06	<0.001	−0.04	0.04	0.01	−0.03	0.04
Step 2									
Gender (0 = M, 1 = F)	−0.10	−0.66	0.47	0.28	−0.27	0.83	0.37	−0.07	0.82
Age	0.01	−0.03	0.04	<0.001	−0.03	0.03	<0.001	−0.03	0.03
Negative Mental Health Attitudes	−0.02	−0.28	0.25	−0.08	−0.33	0.18	0.06	−0.15	0.26
External Shame	0.19	−0.05	0.43	0.15	−0.09	0.38	0.13	−0.06	0.32
Internal Shame	−0.05	−0.35	0.24	−0.11	−0.40	0.18	−0.05	−0.28	0.18
Reflected Shame	0.11	−0.13	0.35	0.08	−0.15	0.32	0.09	−0.10	0.28
Self-Compassion	−4.22 ***	−5.51	−2.94	−2.99 ***	−4.24	−1.74	−3.22 ***	−4.23	−2.22
Caregiver Identity	0.41	−0.17	0.99	0.38	−0.19	0.94	0.04	−0.41	0.50
Adj R^2^ Δ	49%			26%			43%		

B = unstandardised regression coefficient. * *p* < 0.05; *** *p* < 0.001.

**Table 4 healthcare-10-00584-t004:** Multiple regression: mental health shame and caregiver identity to self-compassion among education students (*n* = 109).

	Self-Compassion		
		95% CI	
	B	Lower	Upper
Step 1	1.75	1.54	1.97
Gender (0 = M, 1 = F)	−0.06	−0.16	0.03
Age	<0.001	−0.01	0.01
Step 2	2.02	1.54	2.50
Gender (0 = M, 1 = F)	−0.02	−0.10	0.07
Age	<0.001	<0.001	0.01
Negative Mental Health Attitudes	−0.02	−0.06	0.03
External Shame	−0.04 *	−0.07	<0.001
Internal Shame	−0.07 ***	−0.11	−0.03
Reflected Shame	0.03	−0.01	0.06
Caregiver Identity	−0.03	−0.12	0.06
Adj R^2^ Δ		25%	

B = unstandardised regression coefficient. * *p* < 0.05; *** *p* < 0.001.

## Data Availability

The data that support the findings of this study are available from the corresponding author, Y.K., upon reasonable request. All procedures followed were in accordance with the ethical standards of the responsible committee on human experimentation (institutional and national), and with the Helsinki Declaration of 1975, as revised in 2000 (5). Informed consent was obtained from all patients for being included in the study. No animal or human studies were carried out by the authors for this article.

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
