# Peer review of "Mental Health Shame, Caregiver Identity, and Self-Compassion in UK Education Students"

_healthcare, 2022, doi:10.3390/healthcare10030584_

Round 1

Reviewer 1 Report

The authors aimed in the present study to explore relationships among mental health problems, mental health shame, self-compassion, and caregiver identity in UK education students.

The article reads well, and the topic is relevant. The authors may consider to apply a few changes proposed in the following comments.

The introduction reads well and has been written following a logical information flow making it simple and exhaustive at the same time.

Methods and results sections are clear and well written.

I personally have only one question to address to the authors: since the focus of this study was on college students, please explain whether mental health problem (depression, anxiety and stress) and mental health shame in this particular group is mirroring the whole population and can be generalized or not.

You results are interesting as well as expectable. Also, your conclusions make sense.
Generally, the discussion is well structured, although suggestion for driving furter research should be included

Author Response

Response Letter

Manuscript ID: healthcare-1613425

"Mental Health Shame, Caregiver Identity and Self-Compassion in UK Education Students”

Dear Reviewers,

Thank you for your helpful feedback. We have systematically revised our manuscript addressing the points you have raised. Please see our responses below. We hope this revised paper is now acceptable for publication. We extend our sincere gratitude to you for your feedback that has significantly helped to strengthen the paper.

Reviewer 1

Reviewer 1’s comment 1

The authors aimed in the present study to explore relationships among mental health problems, mental health shame, self-compassion, and caregiver identity in UK education students.

The article reads well, and the topic is relevant. The authors may consider to apply a few changes proposed in the following comments.

The introduction reads well and has been written following a logical information flow making it simple and exhaustive at the same time.

Methods and results sections are clear and well written.

I personally have only one question to address to the authors: since the focus of this study was on college students, please explain whether mental health problem (depression, anxiety and stress) and mental health shame in this particular group is mirroring the whole population and can be generalized or not.

You results are interesting as well as expectable. Also, your conclusions make sense.

Generally, the discussion is well structured, although suggestion for driving furter research should be included.

Authors’ response 1-1

Thank you for your helpful and thoughtful feedback. We are pleased to hear it. Regarding the generalisation to the whole population, as we used referential statistics, it is reasonable to assume our findings would apply to the general UK education student population, however mental health problems and shame are reported differently in different cultures, therefore the results may be different in other cultural groups. This is now added to the manuscript (p.9).

Moreover, additional suggestions for further research are now added (p.9).

Thank you for your insightful suggestions.

Reviewer 2 Report

The publication submitted to me for review analyses issues that are currently significant and relevant.

The content of the article is precisely and clearly structured. The aims are specific and are answered in the content. The content of sections is complete and presented in a consistent manner. The results are analysed and presented appropriately. The discussion is constructive, the conclusions are concrete.

I would recommend rethinking the title of the article so that it is worded without punctuation marks. It is an established practice for the title to be formulated in a single specific statement, sometimes specifying it after the colon.

Subsection 1.1 is entitled “Mental Health Problems”, but the content itself little reveals these problems, this needs to be supplemented.

References are given differently in some places in the article, this needs to be carefully reviewed.

I hope that this article and that it will be successfully published. I wish the authors the best of luck.

Author Response

Response Letter

Manuscript ID: healthcare-1613425

"Mental Health Shame, Caregiver Identity and Self-Compassion in UK Education Students”

Dear Reviewers,

Thank you for your helpful feedback. We have systematically revised our manuscript addressing the points you have raised. Please see our responses below. We hope this revised paper is now acceptable for publication. We extend our sincere gratitude to you for your feedback that has significantly helped to strengthen the paper.

Reviewer 2

Reviewer 2’s comment 1

The publication submitted to me for review analyses issues that are currently significant and relevant.

The content of the article is precisely and clearly structured. The aims are specific and are answered in the content. The content of sections is complete and presented in a consistent manner. The results are analysed and presented appropriately. The discussion is constructive, the conclusions are concrete.

I would recommend rethinking the title of the article so that it is worded without punctuation marks. It is an established practice for the title to be formulated in a single specific statement, sometimes specifying it after the colon.

Subsection 1.1 is entitled “Mental Health Problems”, but the content itself little reveals these problems, this needs to be supplemented.

References are given differently in some places in the article, this needs to be carefully reviewed.

I hope that this article and that it will be successfully published. I wish the authors the best of luck.

Authors’ response 2-1

Thank you for your helpful and thoughtful feedback. In line with your comment, our title of the manuscript and of Subsection 1.1 are revised. References are corrected.

Reviewer 3 Report

I consider that the article reveals itself as very interesting and very relevant. Besides the fact that there are not many studies in this area and taking into account the limitations described by the authors, it will be interesting to pursue studies in this field of study, opening the door to possible international cooperation. 
We consider that the article is well written and suitable for submission. 
We only suggest that on page 8, line 289, the full stop that is after (Kotera, 2021) be removed.

Author Response

Response Letter

Manuscript ID: healthcare-1613425

"Mental Health Shame, Caregiver Identity and Self-Compassion in UK Education Students”

Dear Reviewers,

Thank you for your helpful feedback. We have systematically revised our manuscript addressing the points you have raised. Please see our responses below. We hope this revised paper is now acceptable for publication. We extend our sincere gratitude to you for your feedback that has significantly helped to strengthen the paper.

Reviewer 3

Reviewer 3’s comment 1

I consider that the article reveals itself as very interesting and very relevant. Besides the fact that there are not many studies in this area and taking into account the limitations described by the authors, it will be interesting to pursue studies in this field of study, opening the door to possible international cooperation.

We consider that the article is well written and suitable for submission.

We only suggest that on page 8, line 289, the full stop that is after (Kotera, 2021) be removed.

Authors’ response 3-1

Thank you for your helpful and thoughtful feedback. Now the line 289 is corrected.